# Environmental Factors and Kawasaki Disease Onset in Emilia-Romagna, Italy

**DOI:** 10.3390/ijerph17051529

**Published:** 2020-02-27

**Authors:** Elena Corinaldesi, Valentina Pavan, Laura Andreozzi, Marianna Fabi, Andrea Selvini, Ilaria Frabboni, Paolo Lanzoni, Tiziana Paccagnella, Marcello Lanari

**Affiliations:** 1Pediatric Department, Ramazzini Hospital, Carpi, 41012 Modena, Italy; p.lanzoni@ausl.mo.it; 2Arpae-SIMC, Hydro-Meteo-Climate Service of the Regional Agency for Prevention, Environment and Energy (ARPAE), 40122 Bologna, Italy; vpavan@arpae.it (V.P.); aselvini@arpae.it (A.S.); tpaccagnella@arpae.it (T.P.); 3Pediatric Emergency Unit, Medical and Surgical Sciences Departement, S.Orsola-Malpighi Hospital, University of Bologna, 40138 Bologna, Italy; laurandreozzi@gmail.com (L.A.); marianna.fabi@aosp.bo.it (M.F.); ilaria.frabboni@gmail.com (I.F.); marcello.lanari@unibo.it (M.L.)

**Keywords:** Kawasaki disease, environmental factors, air quality, meteorological parameter

## Abstract

Kawasaki disease (KD)is an idiopathic acute febrile illness that primarily occurs in children <5 years of age and can lead to artery lesions if not promptly treated. Recent studies have shown possible correlations between KD onset and currents and air pollutants.The present study describes results on the correlation between environmental conditions, frequency, and variability ofKD onset in children under five years of age in Emilia-Romagna, a region of Northern Italy, over the period from 2000 to 2017. Since there are substantial climatic differences between the western-central (Emilia) and the eastern area (Romagna) of the region, the data for these areas are analyzed separately. The environmental factors considered are observed local surface daily temperature, daily precipitation, upper air wind regimes, and local air pollution. The results indicate that in Emilia-Romagna, KD onset occurs mainly during late autumn and early spring, which is in agreement with the literature. The frequency of KD onset in Emilia is significantly higher in months characterized by a high frequency of southerly flow, which is associated with milder than average night-time temperature, and in years with a prevailing south-westerly mean flow. These results are consistent with other studies, suggesting that certain wind conditions are more favorable for disease onset, which are possibly associated with one or more airborne agents.

## 1. Introduction

Kawasaki disease (KD) is an acute and self-limited vasculitis that particularly affects children under 5 years, leading to coronary arteries alterations in about 15%−25% of cases if not promptly treated [1]. The disease has been reported worldwide with different incidences in different countries [1], but it is known to be more common in East Asianpopulations with a peak of 309/100,000 cases for children under 5 years of age in Japan [2]. In Europe, the incidence of KD is variable from 5 to 16 cases per 100,000 children under 5 years of age. A recent Italian study [3] reports the incidence under 5 years of age in the Emilia-Romagna region as equal to 16.4 cases over the 2011 to 2013 period, with a peak in winter to early spring and a fall in the summer. The etiology of KD seems to be multifactorial, with some infectious trigger on genetically predisposed children playing an important role. Recent epidemiological studies have focused on some potential environmental risk factors for KD and the analyses have correlated the incidence of KD cases in Japan, Hawaii, and San Diego with tropospheric wind currents originating from northeastern China, which suggests that a wind-borne agent could trigger the illness [4,5].

Burns et al. [6] have studied the seasonal patterns of KD over the globe in the years 1970–2012 in 25 countries. In the Northern Hemisphere extra-tropics, the cases of KD cases were higher from January to March than from August to October, while in the tropics and the Southern Hemisphere extra-tropics, the maximum incidence of KD was between May to June and was lower between February, March, and October. Their results also support the hypothesis that an environmental trigger could have a role in determining the seasonality of KD cases worldwide [7,8,9]. In particular, a recent study has shown that large scale tropospheric wind patterns could be associated with fluctuations in KD cases: Rodò et al. [4,5] hypothesized the existence of a KD trigger agent spread by means of direct airborne sampling conducted over Japan. In 2014, the same authors conducted a detailed analysis of nucleic acids extracted from aerosolized atmospheric samples trapped on filters collected at selected altitudes, finding a different microbiota of the tropospheric aerosols during KD seasons, with Candida as the dominant fungus [5]. Besides, Chilean peaks of incidence have been related with air masses that originated in the northern Atacama desert [10]. All these results are consistent with the presence of an external environmental trigger for KD interplaying with a genetic susceptibility.

Recently, some studies examined the association between KD onset and exposure to specific air pollutants, such as carbon monoxide (CO), nitrogen dioxide (NO_2_), ozone (O_3_), particulate matter PM10 and PM 2,5 and sulfate dioxide (SO_2_) [11,12,13]. No relation was found by Zeft et al. [11] and Lin et al. [12], while Jung et al. [13] found a relation between KD onset frequency and exposure to high level density of O_3_ in Taiwan. Finally, a recent study by Yorifuji et al. [14] showed that prenatal exposure to high levels of particulate matter may significantly increase the risk of KD occurrence in child from 6 to 30 months of age in Japan.

The purpose of the present study is to describe the correlations between KD onsets in Emilia-Romagna, a region of Northern Italy, and the variability of local climate in terms of wind direction and intensity, surface temperature, precipitation and local density of environmental pollutants, like urban and rural air concentration of particulate matter, detected at local air quality, and meteorological stations. This study represents a novel contribution to the field since KD European data has never been correlated with local meteorological or environmental variability except for a recent study conducted in Spain, where no significant associations have been found between KD and regional Weather Types, although a possible relevance of some environmental factors in disease onset has been suggested [15].

## 2. Materials and Methods

### 2.1. Study Sample and Setting

The KD dataset consists of the number of KD onsets in the population served by the Hospitals of Emilia-Romagna, a region of Northern Italy, focusing on children under 5 years. The data were extracted from the regional hospital discharge records database, including all admissions to hospitals for residents in Emilia-Romagna from 1 January 2000 to 31 December 2017. Analyzed data only include admissions for children younger than 5 years of age with KD code ICD9 446.1 as their main discharge diagnosis. This is a limitation of this study because relying on discharge data may lead to an overestimate of the incidence of the disease. For each patient, the calendar day of disease onset is available. The locations of the hospitals are shown on the map in Figure 1. In order to study the correlation between KD onset and environmental data, only data from the hospitals located in the plain areas were considered. Figure 2a presents the density of children under 5 years in each municipality of Emilia-Romagna in 2003, while Figure 2b presents the total number of KD onsets in each hospital over the study period. The information reported in the two maps is consistent with higher numbers of KD occurrences in the hospitals located in more populated areas. Although Figure 2a is relative to a particular year, it is representative over the whole period considered. As can be seenin Figure 2a, a young population is mostly concentrated in the plain area, with the exception of three municipality over the Apennines (Porretta Terme, Pavullo nel Frignano, and Castiglione ne’ Monti). The climate over the Apennines is substantially different from that in the plains and the contribution of the three Apennine Hospitals to the overall statistics is tiny due to the low population density in their area. This is also confirmed by Figure 2b. For this reason, the cases from these three hospitals were not included in the present study so as to focus on the relation between occurrence of KD and local environmental conditions in the plains. Since, as described in Appendix A, the climate in the plains depends substantially on longitude and distance from the Adriatic coast, the hospitals are divided into two groups: those located in the central-western area in Figure 1 (hereafter Emilia) and those located in the eastern area (hereafter Romagna). This allowed us to identify three monthly indices: the number of KD onsets in the whole plain area of Emilia-Romagna, the index of occurrences in Emilia, and that of occurrences in Romagna.

The indices of number of KD onsets have been renormalised to 100,000 children, using the number of patients diagnosed in each year over the considered area, obtained from the ISTAT database (https://www.istat.it/).

### 2.2. Environmental Covariates

Meteorological data include monthly climate indices obtained from local daily cumulated precipitation and daily minimum and maximum surface air temperature at 2 m above the ground over the same period covered by the KD data. These data have been extracted from the surface observational daily analysis, operationally produced by ARPAE-SIMC over Emilia-Romagna on a regular latitude-longitude grid with an approximate 5 km resolution, available from 1961 to present. The analysis includes minimum and maximum daily temperature and daily cumulated precipitation. The methods used to produce the analysis starting from validated local daily observational data are described in Antolini et al. [16]. Wind data have been extracted from the Limited Area Model Analysis (LAMA) dataset, which is an initialized local analysis obtained with the limited area model of the Consortium for Small-scale Modeling (COSMO) combining the operational analysis produced by the European Center for Medium Range Forecasts (ECMWF), as a first guess, and observational Global Telecommunication System (GTS) data over the area. The set covers the period from 2005 to present. The COSMO model, formerly known as Lokal Modell, was initially developed at the Deutscher Wetter Dienst [17] and is currently maintained by the COSMO consortium, involving several European meteorological services including the regional Hydro-Meteo-Climatological Service of Emilia-Romagna (ARPAE-SIMC). The LAMA analysis is characterized by an approximate horizontal resolution of 7 km and 40 vertical levels from surface up to 30 hPa. In the present study, only 12:00 noon UTC (Coordinated Universal Time) instant wind intensities and directions at 850 hPa were used. These data were preferred to observational data at 10 m from the surface because, unlike observational data, they refer to a level located out of the planetary boundary layer and are more representative than surface data of the wind conditions over a large area.

The relation between wind regimes and large scale circulation anomalies is described using monthly mean values of 500 hPa geopotential height (Z500) from the ERA5 reanalysis dataset [18] made available by ECMWF through the Copernicus Services.

The relation between the frequency of KD occurrences and air pollution is described using data of air concentration of particulate matter with mass median diameter less than 10 micrograms (PM10), from a network of observational stations managed by ARPAE from 2002 to present.

### 2.3. Statistical Analyses

Indices describing the climate variability were obtained by averaging grid point values over each area, which are functions of local daily precipitation and minimum and maximum temperature. The climate indices are: total monthly and annual precipitation and monthly 10th, 50th, and 90th percentiles of minimum and maximum temperature (T_min_ and T_max_), together with their anomalies with respect to their monthly climatology and their mean annual values.

As for the wind data, the indices used hereafter include annual and monthly frequencies of occurrence of wind calms (defined as wind intensities smaller than 3 m/s of all directions) and annual and monthly frequencies of the occurrence of specific wind directions associated with wind intensities greater than 3 m/s. Wind directions follow the typical meteorological convention, indicating the provenience of the air, and have been stratified into eight different classes, centered on major compass directions. As in the case of the other surface parameters, two sets of indices have been produced, one for Emilia and the other one for Romagna.

The statistical significance of all correlation values was checked using Monte Carlo techniques [19] by comparing the observed correlation values with those from 1000 synthetic time series artificially obtained by reordering the KD index with a white noise generator. The use of this technique is meant to identify the presence of a statistically significant correlation between two time series in the presence of strong serial auto-correlations, which could introduce an artificial source of correlation between them.

With respect to air quality data, the number of PM10 stations characterized by an observational time series covering at least 20% of the total period is limited, but a comparison of these data with those coming from a denser observational network installed in the last few years confirms that the general urban and rural characteristics of these parameters are captured by the historical stations. It was not possible to study the relation between KD occurrences and the density of individual air pollutants due to the shortness and geographical scarcity of observational data available for these parameters in Emilia-Romagna. As a consequence, PM10 is used as a proxy of general air quality.

In order to classify the different sources of air pollution, PM10 stations are classified depending on their location. Figure 1 presents the map of the PM10 stations, classified as urban stations, which are more influenced by traffic pollution, and those classified as rural stations, describing the background air pollution. Since the disease incidence data all come from hospitals located in the plains, only stations located in that area were considered. Only data from stations active for at least 90% of the period considered were used, discarding all data from recently installed stations. Using the observational daily PM10 data, it was possible to build a PM10 monthly, monthly anomaly, and annual index for rural and urban areas in Emilia and in Romagna and use them to evaluate the variability of air pollution over the different parts of the region.

## 3. Results

Study subjects include a total of 516 patients affected by KD in Emilia-Romagna between 1 January 2000 and 31 December 2017: 329 (71%) patients in Emilia and 187 (29%) in Romagna; 299 were male (58%) and 217 female (43%),and the disease was more prevalent in girls by ≈ 1.3:1. Median age (months) at onset was 34 ± 30 standard deviation (SD) and 113 patients (22%) were younger than 12 months.

Figure 3a shows the number of KD onsets per month, averaged over the period 2000−2017 over Emilia-Romagna, Emilia, and Romagna, normalized to 100,000 children. KD onset of the disease is less likely to occur from May to September. No significant difference can be observed in the mean monthly frequency of KD onsets over the full period in the two subareas of the region.

Figure 3b describes the time series of the number of KD onset per year per 100,000 children in Emilia, Romagna, and Emilia-Romagna as a function of time (years). Mean numbers over the period are close to 15 onsets per year, which is in agreement with the literature, and greater than the mean number of onsets per year in northern European countries [20]. During the period, the number of KD onsets per year changed substantially, with two outbreaks of the disease in 2005 and 2013, when it reached values up to 30 onsets per 100,000 children. Over the considered period, inter-annual variability also had similar amplitude in the two areas, with peaks in the number of onsets occurring in different years depending on the area.

Table 1 and Table 2 show the values of correlation between the KD onset frequency and temperature indices over the two areas. All monthly temperature indices present a statistically significant anti-correlation with the number of onsets of KD per month, due to the presence of an opposite seasonality in the two indices, explaining most of their respective monthly variance: temperatures reach maximum (minimum) values in the summer (winter) when KD is at its minimum (maximum). This anti-correlation disappears when seasonality is removed from both time series (anomalies). The only significant correlation that is still present in the anomaly time series is that between a higher (lower) monthly 90th percentile of minimum temperature (T_min_ 90th p) in Emilia and a higher (lower) number of KD onsets. No significant correlation was found between KD onsets and day-time maximum temperature in Emilia or between KD onsets and temperature (minimum or maximum) in Romagna or between annual indices.

Correlation values between monthly frequency of KD onsets and monthly precipitation or precipitation averaged over the three months ending at the end of the month considered were computed for both areas (values not shown for brevity). No significant correlation was found in all cases, indicating that the KD outbreaks are not sensitive either to the momentarily increase in humidity or to the improvement in air quality connected with precipitation occurrence, nor are they sensitive to the persistence of such conditions up to three months.

Table 3 and Table 4 shows the correlation values between the frequency of KD onsets and wind indices depending on direction and intensity for the two areas. Correlations are computed for the time series of monthly values, for their correspondent monthly anomalies with respect to monthly climate, and for annual mean indices.

In Emilia, significant correlation values between KD indices and monthly wind indices or monthly anomalies of wind indices are only found for the frequency of southerly wind events, which are typically associated with high values of night-time temperatures (high values of T_min_ 90th p). Figure 4 shows the covariance between the monthly wind anomaly index and the Z500 monthly anomalies for the wind calm and for the southerly wind indices for Emilia, both in terms of Z500 anomalies (shaded fields) and associated full fields (contours). As can be seen, a local monthly anomaly of southerly wind in Emilia (b) is related to a change in the general circulation anomalies favoring the extension of the Atlantic tropospheric jet into the Euro-Mediterranean area and a related high frequency of synoptic transients coming from the Atlantic over the area. On the contrary, during months characterized by a high frequency of wind calm days (a), high values of Z500 are more likely to occur. This correlation is consistent with the previously described presence of a significant correlation, at monthly time scales, between KD onset frequency and the T_min_ 90th panomaly index in the same area. The presence of a barely significant correlation between monthly anomalies of KD onset and easterly wind event frequencies might instead be connected to the fact that these last wind regimes are more typical of months when the KD onset is less likely to occur (specifically January, April, and July). The strong seasonality in the variability of the two indices may lead to a borderline significant value of correlation between the two time series.

Finally, the Emilia annual index of KD onset frequency was significantly correlated only with south-westerly wind regimes, indicating a general similarity between the long-term variability of the two time-series, as can be seen in Figure 5b. In particular, it is possible to appreciate that the KD outbreak in the second half of 2013 in this part of the region was associated with particularly high frequencies of south-westerly winds, reaching an overall mean peak in this year. This wind regime is significantly related to general milder night-time temperatures, but has no correlation with day time temperature (T_max_) like the monthly southerly regime.

Figure 5a can help one to appreciate the relevance of the seasonal component of variability of wind regimes frequency, which can be seen in particular in the calm index, presenting a significant correlation with the number of KD onsets at seasonal time scales (5 month running mean), but not at monthly nor at annual time-scales. Calm of wind frequency has a significant seasonal component, with its maximum in summer months when KD is at its minimum. Interestingly, the periods with a persistently reduced number of KD onsets are sometimes characterized by a greater than average persistence in wind calms.

In Romagna, no significant correlation was present at monthly time scales, while at yearly time-scales, only the westerly wind regime frequency presents a significant anti-correlation value with the yearly KD occurrence frequency, with KD frequency decreasing as the frequency of westerly wind increases. These winds are associated with an increase in the 10th percentile of both minimum and maximum temperature, indicating milder temperature conditions, although none of these relations emerged from the analysis presented earlier of the connections between the local temperature indices and KD onset frequency.

Figure 6b presents the time series of these indices together with their five months running mean, showing a clear anti-correlation in the long term variability. In Figure 6a, it is shown that in Romagna, the relation between the frequency of calm of wind and the KD onsets is weaker than in Emilia, which is probably due to a weaker seasonal component of the indices.

Finally, in Table 5 and Table 6, the relation between KD onset frequency and the air quality is investigated, of which the variability is represented by the PM10 monthly indices. These indices are generally indicative of the mean air quality during a specific month, with high PM10 values pointing towards high air pollution. The presence of the correlation for monthly values, monthly anomalies with respect to the index climate, and annual mean values are evaluated. Both rural and urban conditions are considered.

In Emilia, the monthly times series of rural and urban PM10 are both significantly correlated due to the presence of a strong similar seasonality of the two indices. The absence of a correlation between the correspondent monthly anomaly series seems to indicate the absence of a direct relation between the intensity of local air pollution and KD onset frequencies. The presence of a negative, but not statistically significant, relation between the PM10 indices and KD annual indices is related to the fact that the south-westerly wind annual mean index is the only one that is negatively correlated with the PM10: upper air south westerly wind regimes cause higher chance of rainy conditions and lower values of local pollutants in the air. Although this is a general condition during the occurrence of this specific wind regime and a significant relation has been identified between KD onset frequency and the frequency of S-W wind regimes at an annual scale, there is no direct relation between KD onsets and local PM10.

In Romagna, the correlation values seem to indicate a similar connection between KD onset and local air pollution, although less pronounced and with smaller statistical significance than in Emilia.

## 4. Discussion

This study suggests the presence of a correlation between the frequency of KD onsets and environmental factors and shows that KD onset in Emilia and in Romagna have similar general characteristics. The frequency of onsets per 100,000 children observed in Emilia-Romagna is similar to those observed in other European countries, although it seems higher than Northern European incidences [20]. As described in the literature [2,9,20], KD onsets are more likely to occur from October to April, in particular from late autumn to early spring. The lower frequency of KD onsets during summer seems to exclude that the disease is related to an increase in pollen or spores concentrations, like Stemphylium or Alternaria, and to increases in aerosols generated by agriculture linked to nitrate fertilizers distribution, with typical climatological maximum during summer months. Sources for both of these air pollutants are present in the region considered.

During the study period, the number of onsets per year has changed quite substantially from year to year and it is possible to identify at least two outbreaks of the disease, characterized by a number of onsets more than double the average over the other years. Furthermore, analyzing the time series of KD onset frequency, it is shown that the inter-annual variability over the two areas is slightly different with a similar amplitude and peaks of KD onset in different years. This seems to indicate the presence of a different link with some environmental factors in Emilia and in Romagna.

The presence of a significant correlation value in Emilia between the monthly 90th percentile of T_min_ and the KD onset number seems to suggest that KD onset is more likely to occur during particularly mild periods, with the minimum temperature (night-time temperatures) occasionally reaching values significantly warmer than average. This is in agreement with Rypdal et al. [8] who found a correlation between KD clusters onsets and occurrence of circulation anomalies associated with warmer night-time temperatures. No correlation was found between monthly and annual precipitations and KD onsets, as described in literature.

In Emilia, the monthly frequency of KD onsets is shown to be significantly correlated with monthly weather regimes associated with local southerly winds. Southerly wind regimes are sometimes related to the occurrence of Apenninic föhn events, to a greater boundary layer mixing, and, more often than not, to rainy conditions and mild night temperatures. All these conditions favor mixing of otherwise colder surface air parcels with milder upper air ones. In this way, the significant relation of KD onset frequency in Emilia with milder night temperature and southerly local wind regimes are two sides of the same phenomenon.

In the same area, at the annual level, there is a significant correlation between the frequency of KD onsets and south-westerly regimes. At the annual level, the wind index is significantly correlated with milder night temperature. It is possible that the longer averaging process highlights the relevance of this wind direction, which is more related with upper-air large scale conditions prevailing during local southerly wind events. So, in the end, this significant correlation value also points towards the same class of environmental conditions. During these events, the greater air mixing and the greater likeness of rainy conditions lead to lower air pollution values, as suggested by the PM10 index. This results in a general negative, not statistically significant correlation between PM10 yearly index and KD onsets, showing no direct relation between the two indices, in accordance with what was found by Zeft et al. [11] and Lin et al. [12].Conversely, all these different results point towards the possibility that the KD onset is mostly associated with the occurrence of particular weather regimes associated with upper air wind of a specific direction, which can be responsible for the transport of environmental agents, ultimately triggering the disease in susceptible patients, as suggested in the literature [4,7,8,10].

In Romagna, the only significant relation between KD onset and meteorological indices is the anti-correlation with the annual westerly index. This index, which is not directly related to the type of events favoring the KD onsets in Emilia, is related to generally mild conditions in Romagna, both in the day and in the night. The information collected from the environmental indices considered is not sufficient to identify the reason for the presence of such anti-correlation in this area. It is possible that the difficulty in obtaining consistent results in Romagna for all environmental indices is also linked to the paucity of the number of clinical cases in this part of the region, leading to a greater difficulty in identifying statistically significant relations between the disease onset and environmental factors. On the other hand, it is possible that in this area, the relation with the environment is harder to identify for other unknown reasons due to a more complex relation between the environmental trigger of the disease and weather regimes.

The present work has considered only some of the possible factors that could possibly affect KD onset. In particular, in the absence of specific observational data, it was not possible to verify if high concentrations of carbon monoxide (CO), nitrogen dioxide (NO_2_), ozone (O_3_), PM 2,5, and sulfate dioxide (SO_2_) favor the occurrence of KD onset as described in the literature [11,12,13]. Additionally, no data was available to confirm a recent study by Yorifuji et al. [14], which showed that prenatal exposure to high levels of particulate matter may significantly increase the risk of KD occurrence.

This study has several limitations. Firstly, the small number of cases related to the low incidence of KD in our region and, more generally, in Europe. Secondly, we considered the exposure at hospital location rather than individual exposure. Despite these limitations, the results seem encouraging and encourage an extension of this study to other regions in Italy and in Europe and a more detailed investigation of the correlation between climate factors and possible environmental triggers in air samples, such as spores and microorganisms.

## 5. Conclusions

This study represents a novel contribution to the investigation of the possible correlation between KD onset and climate factors in Europe, including wind patterns and air pollutants.

According to our results, in Emilia, KD onsets are more likely to occur in periods characterized by weather events with substantially warmer night-time temperatures, as previously described [8]. This finding can be related to the significant correlation found between southerly winds (on a monthly scale) and south-westerly winds (on an annual scale) in Emilia, which are associated with the occurrence of warmer night-time temperatures.

Conversely, it is not possible to obtain consistent results in Romagna, probably due to the paucity of the number of clinical cases in this part of the region.

Our study supports the hypothesis of an environmental agent carried by the wind from a specific direction that can trigger KD in patients who are genetically susceptible. No correlations between PM10 index and KD onsets were found.

This is the first study in Europe that is consistent with a possible correlation between KD and wind-born agents. Further investigation is needed to find environmental trigger agents and to confirm a similar relation between KD onsets and climate anomalies in other Italian regions. Collaboration between European nations may help to increase the number of KD cases included and to compare results for regions with different climate conditions and variabilities.

## Figures and Tables

**Figure 1 ijerph-17-01529-f001:**
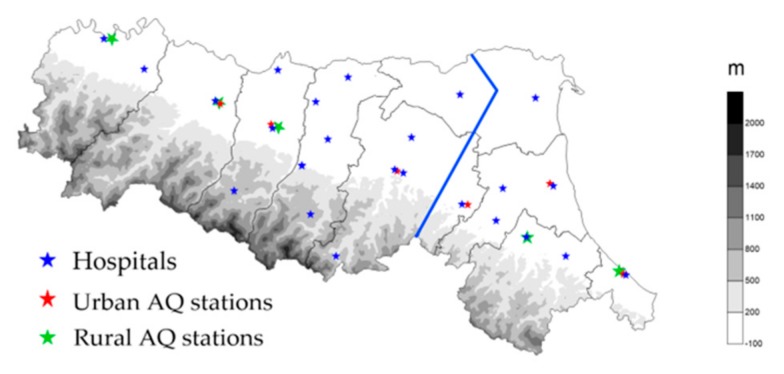
Map of the Emilia-Romagna Region. The blue stars indicate the location of Emilia-Romagna Hospitals, red stars mark the location of urban air quality stations, while green stars indicate the location of rural air quality stations. On the right is the Adriatic sea, on the south west is the Tyrrhenian sea, while the Po river flows from west to east parallel to the northern border of the region. The thick blue line is the border between Emilia and Romagna.

**Figure 2 ijerph-17-01529-f002:**
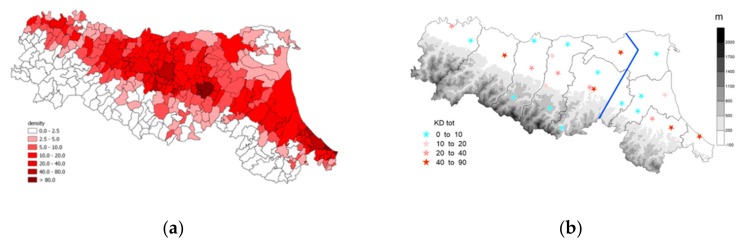
(**a**) Map of density of kids under 5 years of age per km^2^ in 2013 for each municipality of Emilia-Romagna; (**b**) map of hospitals in Emilia-Romagna. Symbols are colored depending on the total number of Kawasaki disease (KD) cases from 2001 to 2017.

**Figure 3 ijerph-17-01529-f003:**
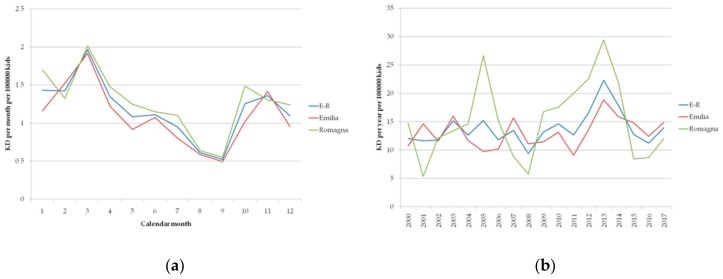
Number of cases per 100,000 kids of reported KD onset in children under 5 years old over the period 2000–2017 (**a**) mean value over the period as a function of calendar month and (**b**) total value as a function of the year in Emilia-Romagna (blue curve) in Emilia (red curve) and in Romagna (green curve).

**Figure 4 ijerph-17-01529-f004:**
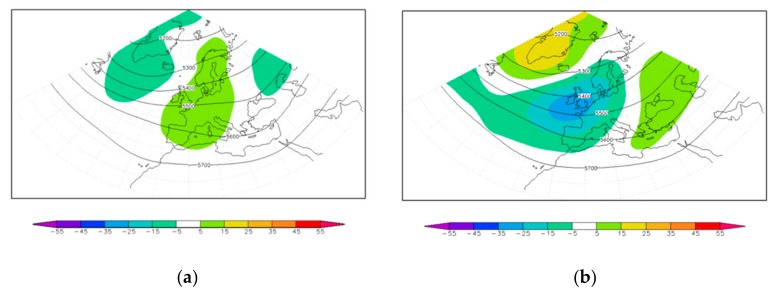
Covariance maps of the Z500 anomalies with the wind monthly anomalies index for the Calm regime (**a**) and for the S wind regime (**b**) over Emilia. Shading shows the covariance pattern, while contours (every 100 m) describe the full field associated with them.

**Figure 5 ijerph-17-01529-f005:**
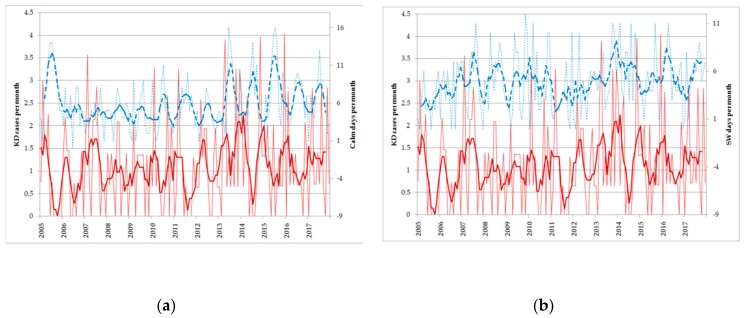
(**a**) Number of KD onsets per month as a function of time in Emilia (thin red line, left axis) together with its five month running mean (thick red line), and number of calm days per month in Emilia (thin dashed blue line, right axis) with its five month running mean (thick dashed blue line). (**b**) Same as panel a but blue lines refer to the number of SW wind per month in Emilia.

**Figure 6 ijerph-17-01529-f006:**
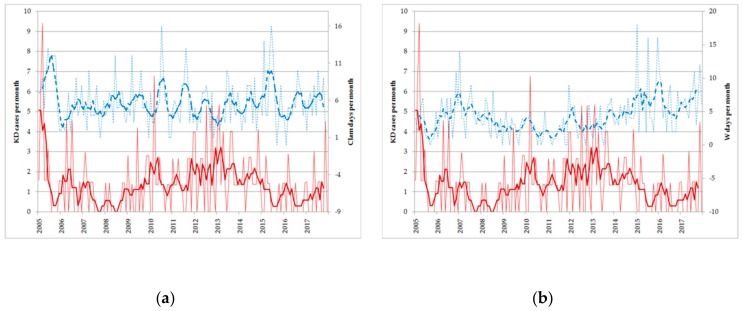
(**a**) Number of KD onset per month in Romagna as a function of time (red thin line, left axis) and its 5 five month running mean (thick red line, left axis) and number of calm days per month in Romagna (thin dashed blue line, right axis) together with its five month running mean (thick dashed blue line). (**b**) Same as panel a, but blue lines refer to W wind days per month in Romagna.

**Table 1 ijerph-17-01529-t001:** Correlation of the monthly time series of KD onsets in Emilia with local monthly climate indices of 10th, 50th, and 90th percentile of minimum and maximum temperature obtained from full fields and anomalies and for their annual mean values.

	Tmin 10th p	Tmin 50th p	Tmin 90th p	Tmax 10th p	T_max_ 50th p	T_max_ 90th p
**Monthly correlation**	−0.23 **	−0.24 **	−0.22 **	−0.22 **	−0.21 **	−0.19 **
**Monthly anomaly**	0.10	0.03	0.13 *	0.10	0.11	0.13
**Annual**	0.33	0.27	−0.01	0.12	0.19	0.04

* indicates statistical significance at the 95% level, ** indicate statistical significance at the 99% level.

**Table 2 ijerph-17-01529-t002:** Same as Table 1 for Romagna. One star indicates statistical significance at the 95% level, while two stars indicate statistical significance at the 99% level.

	Tmin 10th p	T_min_ 50th p	T_min_ 90th p	T_max_ 10th p	T_max_ 50th p	T_max_ 90th p
**Monthly correlation**	−0.16 *	−0.14 *	−0.15 *	−0.18 **	−0.14 *	−0.15 *
**Monthly anomaly**	−0.02	0.06	0.04	−0.10	0.08	−0.001
**Annual**	−0.08	0.08	0.07	−0.40	−0.30	−0.20

* indicates statistical significance at the 95% level, ** indicate statistical significance at the 99% level.

**Table 3 ijerph-17-01529-t003:** Correlation values between the monthly and annual time series of the number of KD onsets and the monthly or annual number of occurrences of one of the nine wind directions and intensity regimes over Emilia. N, N-E, E, S-E, S, S-W, W, N-W stand for the frequency of the nine wind compass direction regimes, while Calm stands for calm day frequency. "Monthly anomalies" indicate the correlation between the time series removed of their seasonality.

	N	N-E	E	S-E	S	S-W	W	N-W	Calm
**Monthly**	0.08	−0.06	−0.13	−0.13	0.18 *	0.06	0.15	0.05	−0.13
**Monthly anomalies**	0.07	−0.13	−0.17 *	−0.14	0.28 **	0.10	0.08	0.01	−0.01
**Annual**	−0.08	−0.42	−0.44	−0.45	0.11	0.62 *	0.33	0.22	0.11

* indicates statistical significance at the 95% level, ** indicate statistical significance at the 99% level.

**Table 4 ijerph-17-01529-t004:** As Table 3 but for Romagna.

	N	N-E	E	S-E	S	S-W	W	N-W	Calm
**Monthly**	0.05	0.04	0.01	0.03	−0.02	−0.09	−0.02	0.04	−0.08
**Monthly anomalies**	0.08	0.04	−0.05	−0.01	0.02	−0.06	−0.06	−0.01	−0.04
**Annual**	0.40	0.36	0.16	−0.13	0.46	−0.23	−0.56 *	−0.19	0.17

* indicates statistical significance at the 95% level

**Table 5 ijerph-17-01529-t005:** Correlation values between the monthly and annual time series of the number of KD onsets and PM10 mean concentrations (μg) in rural (R) and urban (U) stations located in the plains in Emilia (a) and in Romagna.

	PM10 R	PM10 U
**Monthly**	0.20 **	0.20 **
**Monthly anomalies**	0.05	0.04
**Annual**	−0.25	−0.32
**Annualdetrended**	0.06	−0.01

** indicate statistical significance at the 99% level.

**Table 6 ijerph-17-01529-t006:** "Annual detrended" indicates the correlation between the time annual detrended time series.

	PM10 R	PM10 U
**Monthly**	0.09	0.18 **
**Monthly anomalies**	−0.05	0.06
**Annual**	−0.21	−0.11
**Annual detrended**	−0.39	−0.12

** indicate statistical significance at the 99% level.

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
