# Peer review of "Environmental Factors and Kawasaki Disease Onset in Emilia-Romagna, Italy"

_ijerph, 2020, doi:10.3390/ijerph17051529_

Round 1
Reviewer 1 Report
Improved overall.
The yearly outbreaks could be highlighted in the other data better as one would suppose that 2013 would be the highest Southernly winds in the region?
Overall, consider debulking the conclusions/discussion as there is much redundancy (may want to ask editors preference for this journal.
There were a number of poorly worded sentences, a thorough reread would be useful (I listed things I caught).
Line 70, space before “in Taiwan”
Lines- 77-82 is a run on-
Lines 92-94 is poorly worded
Lines 213-221- spacing is askew, same 273 to 275
Table 3- no explanation of what ** means- significant? Not significant?
Line 486. "It is the first study in Europe confirming a possible correlation between KD and wind-born agents."confirming a possible correlation? ‘consistent with a possible’ would be preferred, sounds odd to confirm the possibility of a thing
Author Response
Dear Reviewer,
>The yearly outbreaks could be highlighted in the other data better as one >would suppose that 2013 would be the highest Southernly winds in the region?
The occurrence of an outbreak in the second half of 2013 corresponds to peak values in the frequency of SW wind frequencies as can be seen in figure 5b. A comment on this has been included when descibing this figure.
>Overall, consider debulking the conclusions/discussion as there is much >redundancy (may want to ask editors preference for this journal.
>There were a number of poorly worded sentences, a thorough reread would be >useful (I listed things I caught).
>Line 70, space before “in Taiwan”
Done
>Lines- 77-82 is a run on-
>Lines 92-94 is poorly worded
>Lines 213-221- spacing is askew, same 273 to 275
Corrected
>Table 3- no explanation of what ** means- significant? Not significant?
A sentence has been added both in Table 2 and Table 3 caption, specifying the stars have the same meaning as in Table 1. The fact that annual correlations with greater value than monthly ones are less significant than those depends on the number of degree of freedom (numerosity of the statistical sample) of the time series.
>Line 486. "It is the first study in Europe confirming a possible correlation >between KD and wind-born agents."confirming a possible correlation? >‘consistent with a possible’ would be preferred, sounds odd to confirm the >possibility of a thing
Done
Reviewer 2 Report
Authors have addressed most comments by reviewers.
I strongly suggest to add a limitations section at the end of the discussion addressing the different points raised by reviewers concerning the study which might be a source of bias in results. Specifically exposure attribution at hospital location rather than individual exposure, limited spatial resolution of data and the choice of excluding mountainous areas and other raised by other reviewers.
Figures 4-5 . probably the comment raised was not clear. Having the two trends that are currently overlapping makes the figures hard to read and not very informative. In order to better appreciate the trends in the two data fields shown, if 1 line is shifted upwards (just by changing the range of the y-axis) it is easier to read and we can appreciate the trend and any common/opposite fluctuations as authors themselves suggest.
Author Response
Dear reviewer,
please find in the following our answers:
>I strongly suggest to add a limitations section at the end of the discussion >addressing the different points raised by reviewers concerning the study which >might be a source of bias in results. Specifically exposure attribution at hospital >location rather than individual exposure, limited spatial resolution of data and the >choice of excluding mountainous areas and other raised by other reviewers.
We added a limitations section at the end.
>Figures 4-5. probably the comment raised was not clear. Having the two trends >that are currently overlapping makes the figures hard to read and not very >informative. In order to better appreciate the trends in the two data fields >shown, if 1 line is shifted upwards (just by changing the range of the y-axis) it is >easier to read and we can appreciate the trend and any common/opposite >fluctuations as authors themselves suggest.
Done.
Reviewer 3 Report
The study is in much better shape, particularly the methodology section that seems to be shortened and rewritten for greater clarity. I think it should be published and would be a significant contribution to the field of Kawasaki disease (KD) and the search for its causes. However, I still have a few minor changes and additions requested for the paper before it proceeds out of peer-review:
Revision #1
The abstract requires an introductory sentence or two that gives background on KD in general and also why this study was conducted in the first place:
- A description of KD as an idiopathic acute febrile illness primarily occurring in children <5 years of age would be helpful, or that it is an autoimmune vasculitis causing coronary artery lesions (the most important fact about KD and why we urgently diagnosis it so we treat patients with IVIG 5-10 days after onset).
- Afterwards, please reference that previous epidemiological studies have reported associations for KD incidence with wind currents and air pollutants (yeast, particulate matter) – which led to the current study first being conceived and then performed.
It may lengthen the abstract, but I feel this is necessary particularly to help foster the type of interdisciplinary interest and collaboration that is required for us to identify the cause(s) of KD between climate and environmental researchers and those who study KD.
Revision #2
The first paragraph of the introduction should revise “Asiatic populations” to “East Asian populations”
This is very important as it is more accurate / specific (includes Japan, Korea, Taiwan, China with by far the highest incidences in children <5 years). In addition, no one has established the actual incidence in Southeast Asian countries where resources are limited and differential diagnoses for KD are much more prevalent, endemic – particularly measles. The prevalence of KD also differs in South Asians, while being also not being clearly defined in Central Asia and the Middle East. Therefore, it is important not to generalize the Asian population or KD epidemiology like this. All of this background does not need to be included, however.
Revision #3
I think that the methodology section still requires some subheadings as it is rather detailed and could benefit from some clarity. I would suggest the following formatting for subheadings to help the reader:
- Study Sample and Setting (for KD patients and studied region)
- Environmental Covariates (for climate, environmental agent data description and sources)
- Statistical Analyses (all statistical analyses should be confined to this section)
Revision #4
There are many sentence breaks in the study. Many parts of the paper have single sentences that stand alone. These should be combined into paragraphs wherever possible with only a few left that are deemed necessary by the authors. In particular, this is a bit disruptive for the reader during the conclusion section.
Revision #5
The paper is well-written but I have a few minor spelling changes:
- last paragraph of the discussion:
- change “… and sulfate dioxide (SO2) favous the occurrence…” to “favors”
- last paragraph of the discussion:
- change “No data was either available…” to “Furthermore/Additionally, no data was available…”
Revision #6 (optional)
If possible, the discussion section would benefit greatly from the authors discussing prospective study that could be conducted in this population or others. This is also an opportunity for the authors to elaborate more on their own ideas about this topic, in addition to including their own recommendations for further research on the subject. I would include this after the discussion of the study limitations for your current study, so it expands on how this could be overcome.

Author Response
Dear Reviewer,
thank you for all your comments;
>Revision #1
We have modified the abstract following your suggestions.
Revision #2
We corrected with East Asian populations the prevoius Asian populations.
Revision #3
Done
Revision #4
Done
Revision #5
Done
Revision #6
We added at the end of the discussion the limitations of our study and future prospectives of research.
This manuscript is a resubmission of an earlier submission. The following is a list of the peer review reports and author responses from that submission.
Round 1
Reviewer 1 Report
Overall the data they show with outbreaks particularly in Romagna undercut their conclusions that the wind patterns correlate with cases, as that region does not. The authors should show more detailed analyses for the total dataset (I am suspicious that that analyses will not show any correlations). The authors go through a long explanation of how the regions are geographically distinct which is excessive. The papers sited on Japan and SD do not draw such geographic separations done here.
The section explaining tables 1-3 are poorly constructed. They are redundant of the table legends in most cases.
There are notable limitations in this study, in particular relying on discharge billing data may overestimated your incidence by 20-30 % (see recent papers). This may undermine some conclusions and may explain some of the regional disparity.
There are numerous spacing issues. They are missing Reference 17 (although I am not sure one reference should be used to refer to Northern Europe KD incidence). There are numerous other references that they could include particularly when discussion outbreaks, clustering, and limitations.
The introduction needs extensive revision. I would initially eliminate all semicolons. There are numerous run on sentences that are confusing to the reader. Anti-correlation is usually referred to as negative correlation.
Reviewer 2 Report
Review of Environmental Factors and Kawasaki disease onset in
Emilia-Romagna, Italy: a possible correlation.
Kawasaki disease is a pediatric illness of unknown cause. Its incidence has a distinct seasonal pattern in most locations around the world, but we do not know what causes this feature. Some studies have suggested that Kawasaki disease may have environmental triggers. The authors present an interestig analysis where they correlate Kawasaki incidence data and time series for several climatic and environmental quantities. The data are not conclusive, and the authors fully appreciate this. However, studies of this type are essential for making progress in Kawaski disease research.
As a reviewer, I have no significant issues with the methodology, discussion, or conclusions of the paper. Unfortunately, the paper looks to be a bit unfinished. There are a few "sloppy" things. For instance, line numbers are missing from most of the document, and there are two captions for Table 2. Some text is crossed out.
My recomendation is a major revision that focuses on the polishing the paper to make it publication-worthy.
Below are some specific remarks:
I don't like the title. What about "Environmental Factors and Kawasaki disease onset in Emilia-Romagna, Italy"? I suggest the following re-write of the abstract: "We present results on the correlation between environmental conditions, frequency, and variability of Kawasaki disease (KD) onsets in children under five years of age in Emilia-Romagna, a region of Northern Italy, over the period from 2000 to 2017. Since there are substantial climatic differences between the western-central (Emilia) and the eastern area of the region (Romagna), the data for these regions were analyzed separately. The environmental factors considered are observed local surface daily temperature, daily precipitation, upper air wind regimes, and variability of local air pollution. The results indicate that in Emilia-Romagna, KD onsets occur mainly during late autumn and early spring, in agreement with the literature. The frequency of KD onsets in Emilia is significantly higher in months characterized by a high frequency of southerly flow, associated with milder than average night-time temperature and with south-westerly mean flow. These results are consistent with other studies that suggest that certain wind conditions are more favorable for disease onset, possibly associated with one or more airborne agents." Generally, the authors overuse colons and semicolons. My suggestion is to remove them or them replace these with periods, consistently throughout the text. The authors state that the statistical significance of all correlation values has been estimated from Monte Carlo techniques using synthetic time series obtained of a Gaussian noise generator. I assume that they mean a white (uncorrelated) noise. The problem with this approach is that climatic variables generally have strong serial correlations. One should expect the authors to discuss this problem, at least. Figure 3b: It is not necessary to write "Time (yr)" on the x-axis. I don't understand why there is a dark gray color in the background of all the figures. The tables could be formatted to look better. Remove lines and align to the right, maybe? Use fonts consistent with the rest of the text.
Reviewer 3 Report
Introduction can be improved to better set the scene and the rationale of the study.
Aslo the aim can be improved suggesting th added value of the study and mentioning the weather data.
Please refer to air quality data or particulate matter (PM9 as this is the only "environmental data"" considered.
Methods
There seems to be some confusion regarding exposure attribution.
Hospitals have a rather large catchment area as shown in map 1stretching from the hills to the plains. Potentially subjects being hospitalized are exposed to different meteorological conditions based on area of residency and not all exposed to weather at hospital location it is the ned point where they go in response to insurgence of disease or health condition. It is not clear whether exposure is for the hospital location or attributed to each subject based on residency which can be denoted from hospital discharge data. This would be the most correct way of attributing exposure and considering the limited number of cases it should not be a problem to be done this way. This would ensure no bias in exposure and is attributed. Given the high resolution weather data this shouldn’t be an issue as well.
Hospital exclusion. Please indicate which hospitals were excluded. However we suggest to include all hospitals as sensitivity analysis. Hospital may be wider catchment areas of patients also from pre-alpine areas. So unless you initially select HA only from patients living in the plain low lying areas you are also including subjects from other areas exposed to slightly different conditions. So a sensitivity analysis is encouraged here. Either all hospitals or a sub selection of patients only of residents plains.
Hospital discharge KD diagnosis in any position or main cause? Please specify and include frequency for each year.
Please include which weather parameters were used in previous studies to justify the choice made here.
Make one figure of maps 1 and 2, with different symbols as it would be useful to see where these are located in the same map.
I suggest to carry out a weighted mean by population location rather than giving the same weight to each grid square in the area, as presumably the population is exposed to conditions in areas where they live. So areas or grids with no or very low populations should contribute less in the average.
Page 3 Please change “environmental pollution” with “air pollution” or “air quality”
PM monitoring distribution is heterogeneous some provinces do not have monitors, what was done in those cases? Mention this in limitations section. Would modelled data not have been a possible alternative? Or again if residency location exposure was considered was other data available?
Please include sub-heading materials\data and methodsPlease expand on the statistical analysis carried out and correlation. This is weak. Potentially more sophisticated statistical analysis could be carried out here, such as regression analyses for example. Please include this in the discussion as potential further study.
Last paragraph methods
Not clear needs rewording. Again here the exposure initial issue is important as mentioned at the beginning. Subjects are exposed to conditions at residency not at hospital location as they are hospitalized as a consequence of their exposure.
Possible rewording:
Given the differences in weather regimes and environmental conditions over the two considered areas, mainly due to the different distance from the sea and to the relation between orography and the meteorological flow, it is crucial to carry out separate analyses for each area in order to better characterize the association of local environmental factors on KD onset.
Results
Results page 6 “…35 onsets a year” figure 3b does not show this number can you check it. Or clarify what it is referred to is it the whole region?
Figure 3. add description in the text of the differences in Emilia and Romagna.
Considering data availability, results and evidence in the literature it might be worth including both wind speed and direction even as wind vector components to better describe correlation between KD and environmental exposures.
Figure 4-5 if data on winds is not available for 2000-04 please exclude from figures as not informative.
In figure 4 change labels SW and calm “days” not “cases” like in fig. 5
Figure 4-5 . Might be worth separating the two time series one further up (y-axis on the right can be changed in terms of data interval in order to shift it upwards) in order to make them more understandable.
Table 2 please change “Prec” to “precipitation”
Considering results and KD event distribution it would have been interesting to study extreme events as well eg. Romagna 2005, 2013.
Furthermore, instead of considering only wind and temperature air masses and a synoptic approach could be worth considering as an approach especially as there is a seasonal pattern in KD onset and seasonal weather characteristics. It seems that weather in a more holistic sense can be associated with KD onset, especially considering the seasonal pattern of onset and annual variability too.
Discussion
Please include other possible factors affecting KD onset.
Discussion of results should be more linked with other studies in the literature.
Add a limitations paragraph
Check numbering of references in text and in reference list no. 1 is included twice. So not sure if numbering in text is correct.
Check grammar and language throughout, there are corrections (words cancelled but still in the text) and double spacings to correct.
Reviewer 4 Report
This is an important study that investigates possible environmental factors related to Kawasaki disease (KD) onset for the first time in Europe in relation to wind patterns and air pollutants. My suggestion is to publish after some minor reviews, which mainly focus on additional reporting of previous related studies or some of the current study's limitations
There are a few other pertinent studies for the authors to include and cover in the introduction and discussion. This includes three studies finding a positive correlation with air pollutants and two others that yielded no associations. Similar to the current study in Italy -- a previous study in Taiwan found no relation between KD onset and PM10
Intrauterine and Early Postnatal Exposure to Particulate Air Pollution and Kawasaki Disease: A Nationwide Longitudinal Survey in Japan.https://www.ncbi.nlm.nih.gov/pubmed/29212623 Ambient Air Pollutant Exposures and Hospitalization for Kawasaki Disease in Taiwan: A Case-Crossover Study (2000-2010).
https://www.ncbi.nlm.nih.gov/pubmed/27458717 Association of Kawasaki disease with tropospheric winds in Central Chile: is wind-borne desert dust a risk factor?
https://www.ncbi.nlm.nih.gov/pubmed/25743034 Ambient air pollution, temperature and kawasaki disease in Shanghai, China.
https://www.ncbi.nlm.nih.gov/pubmed/28822259 Kawasaki Disease and Exposure to Fine Particulate Air Pollution.
https://www.ncbi.nlm.nih.gov/pubmed/27496266
As noted by Manlhoit et al. (2018) that is cited by the current Italian study, another factor that may make it harder to identify potential environmental agents in areas such as Northern Italy is the proportion of the population that is genetically susceptible to KD. It may be much easier to identify a potential causative agent in East Asian countries such as Japan where the incidence of KD is by far higher, as Rodo et al. (2014) have previously done with Candida albicans. Collaboration between European nations may help overcome this limitation and make such examinations more robust by increasing the number of KD cases included, in addition to surveying numerous regions with differing climate conditions and variabilities
On this note, the authors should discuss a previous study in Spain by Riancho-Zarrabeitia et al. (2018) that looked at weather types across the country but found no associations. They suggest that the lack of association should be approached with caution, as there is a low incidence in the country (12 cases per 100,000 children <5 years) that makes it more difficult to perform robust statistical analyses. In addition, they did not look at local weather patterns and suggest analyses between regions -- as the current Italian study has conducted for Emilia and Romagna
Kawasaki disease in Spanish paediatric population and synoptic weather types: an observational study.https://www.ncbi.nlm.nih.gov/pubmed/29846788
Furthermore, the authors only looked at PM10 as a general indicator of air pollution level but did not examine individual air pollutants as other studies have done. In addition, intrauterine exposure was not taken into account. Another study cited above in Japan by Yorifuji et al. (2018) suggests higher exposure rates to suspended particulate matter <7 µm in children is associated with KD risk -- particularly in utero
Lastly, the authors should cite the most recent epidemiological survey of KD incidence in Japan, which currently stands at 309 cases per 100,000 children <5 years
Nationwide epidemiologic survey of Kawasaki disease in Japan, 2015-2016.https://www.ncbi.nlm.nih.gov/pubmed/30786118
Also, if possible, the background of the figures should be made a bit lighter. The dark grey background of the figures make them somewhat difficult to examine